# Fracture of the Bone Inducing Its Necrosis as the End Point in the Evolution of Untreated Neuroarthropathy

**DOI:** 10.3390/medicina58010011

**Published:** 2021-12-22

**Authors:** Dured Dardari, Alfred Penfornis

**Affiliations:** 1Diabetology Department, Centre Hopitalier Sud Francilien, 91100 Corbeil-Essonnes, France; alfred.penfornis@chsf.fr; 2LBEPS, IRBA, Université Paris Saclay, 91025 Evry, France; 3Paris-Sud Medical School, Paris-Saclay University, 94270 Le Kremlin-Bicêtre, France

**Keywords:** diabetes, Charcot neuroarthropathy, tarsus necrosis

## Abstract

We describe here the case of a female patient with type I diabetes who developed active Charcot neuroarthropathy in the foot. Due to therapeutic noncompliance, talus necrosis was discovered 2 years after the presentation of neuroarthropathy. The impact of untreated neuroarthropathy on the bone is commonly described as fracture and joint dislocation, but we describe the complete disappearance of the bony structure and its necrosis associated with active neuroarthropathy in a patient who refused offloading.

## 1. Introduction

Neuroarthropathy or “Charcot neuroarthropathy” (CN) is a chronic, devastating, and destructive disease of the bone structure and joints in patients with neuropathy; it is characterized by painful or painless bone and joint destruction in limbs that have lost sensory innervation. The physiopathology of this disorder is poorly understood. The complication contributes to multiple fractures and increases the risk of amputation of the affected joint. In the current state of knowledge, the bone affected by CN often stops being described after the fracture occurs, but the focus here is the post-fracture state of the bone before amputation. The common cause of amputation after this complication is often described as a consequence of the onset of ulceration, but a breakdown of the bone structure after its necrosis has never been described. it is important to shed light once more on the serious consequences of foot amputation on the quality of life of patients, with mechanical impact recognized even on the lateral foot [1].

## 2. Patients and Methods

Our 25-year-old patient was treated for type 1 diabetes, which was diagnosed at age 9. She had no other medical history, although she was hospitalized in 2017 to manage severe glycemic imbalance with HbA1c 12%. She measured 167 cm and weighed 62 kg. Clinical examination revealed peripheral neuropathy with paresthesia in the lower limbs and pathological results in the monofilament and the tuning fork tests. Laboratory tests showed microalbuminuria (289 mg/L, normal range <20 mg/L) with a normal glomerular filtration rate (80 mL/min/1.73 m^2^), suggesting incipient nephropathy. Her only treatment consisted of insulin (via a subcutaneous insulin pump). Two months later, she tested positive for an unplanned pregnancy (~6 weeks). Her HbA1c was 9%, and intensive glycemic management was initiated. At 24 weeks pregnant, the patient was hospitalized for vomiting and suspected pregnancy hypertension. Her HbA1c was 7.4% (60% reduction over 6 months), and she weighed 72.6 kg.

The edema around her right ankle and foot was observed (Figure 1). Thrombophlebitis was excluded by venous Doppler ultrasound. X-rays showed no fractures but Chopart dislocation, highly evocative of acute Charcot foot. An ankle ultrasound did not reveal a ruptured ligament. An Aircast^®^ pneumatic boot was prescribed to reduce weight bearing on the foot; the temperature of the foot was 3 °C higher than the contralateral joint. Ten weeks later, the patient underwent an emergency caesarean at 34 weeks due to irregular fetal cardiac rhythm (HbA1c 7.1%).

Magnetic resonance imaging (Figure 2) performed 2 months after childbirth for persistent foot edema yielded a typical image of active CN in the midtarsal, subtalar, and talonavicular joints, with a displaced navicular fracture, talonavicular dislocation, and cuboid fracture. The patient refused our recommendations of complete offloading with a plaster cast or offloading using a splint with patella support.

Three months after the diagnosis of CN, the patient declared that she was actively walking and not offloading (Aircast^®^) her foot joint. She no longer contacted the Diabetology or Orthopedics Department at our hospital. However, she contacted the Diabetic Foot Unit 14 months later because her foot had become completely deformed (Figure 3). She could no longer wear footwear due to the swelling. During the previous 14 months, the patient had not applied any means of offloading. Computed tomography (CT) scan (Figure 4) revealed osteonecrosis of the talus with almost complete destruction of the dome and posterior process, osteonecrosis of the medial cuneiform, and subchondral osteolysis in the tibial pestle, medial malleolus, calcaneus, cuboid, navicular, lateral, and intermediate cuneiforms. The patient had no skin lesions, but a persistent temperature difference between her affected foot and the contralateral joint (>3 °C) proved that CN was still active. She experienced no pain in the affected foot.

## 3. Discussion

CN is a non-infectious disease involving the rapid destruction of osteoarticular structures, always occurring in the setting of neuropathy. Its physiopathology is poorly understood, while knowledge of its causality remains at the theoretical stage. The most common hypothesis based on inflammatory arthropathy was proposed by Jeffcoate [2], who described CN as an increased inflammatory response to a lesion inducing increased bone lysis with the involvement of bone molding factors, specifically the receptor activator of nuclear factor-B ligand and its natural antagonist, osteoprotegerin.

Generally, CN presents two clinical phases: (i) acute and (ii) chronic. The typical clinical picture of acute CN is a red swollen joint with a temperature difference >2 °C compared to the unaffected joint. These symptoms may go unnoticed because of their frequent association with small fiber neuropathy of the lower limbs, the absence of pain, or the disproportion of foot lesions for diagnosis [3]. The most common anatomic description was proposed by Eichenholtz [4] based on clinical and radiological signs. Nevertheless, the anatomical evolution of CN toward the onset of bone necrosis is not cited in Eichenholtz’s classification [4], even though it is widely used to describe CN.

Every therapy implemented for CN has a common basis: offloading. This case report describes the disastrous evolution of CN, initially triggered in two joints. However, the absence of pain and the presence of another site of CN suggest that the CN joint left untreated after at least one round of offloading tends toward necrosis. The dramatic evolution of the case studied here describes the consequences of CN, notably deformation and wounds.

Because of therapeutic noncompliance with a lack of offloading, the talus bone became necrotic, with the bony structure disappearing along with the bone signal on the CT scan. The disappearing bone structure, though tragic for this young woman, is fortunately rare. Samarasinghe et al. suggested a diagnosis of avascular (aseptic) necrosis of the talus instead of CN, which is defined as a group of diseases with bone necrosis as the common denominator [5]. They usually appear in the epiphyses and carpal and tarsal bones, pathology also accompanied by constant pain, often during a growth period and mostly at skeletal points subject to particular stress fracture. Müller–Weiss disease [6] and Freiberg disease [7] can be excluded as differential diagnoses in our case due to the absence of pain, patient age, bone necrosis location, and abstract notion of trauma-inducing stress fractures.

The long duration of active CN caused by the lack of offloading most likely induced an increased level of receptor activator of nuclear factor-B ligand and osteoprotegerin to control the osteoclast differentiation and survival, thus leading to bone necrosis. This was previously described as the mechanism inducing osteoporosis in menopausal women [8], although in the literature we describe [9], rather bone resorption and not bone necrosis when a CN has not been treated.

To conclude, active CN left untreated by offloading will inevitably evolve toward the necrosis of bones.

## Figures and Tables

**Figure 1 medicina-58-00011-f001:**
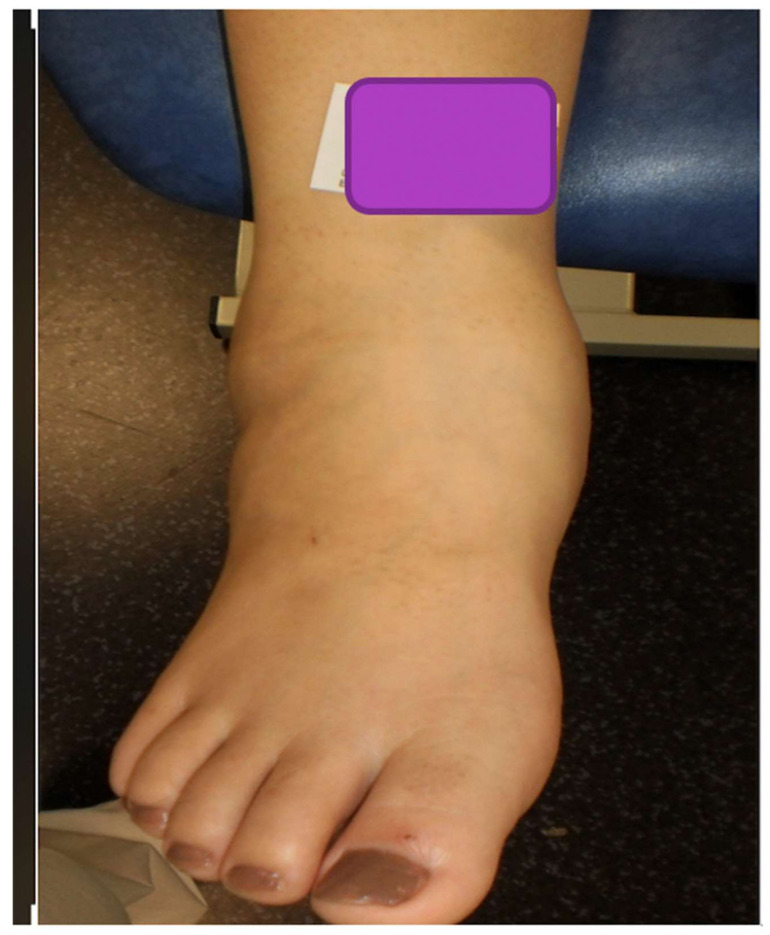
Clinical presentation of Charcot neuroarthropathy.

**Figure 2 medicina-58-00011-f002:**
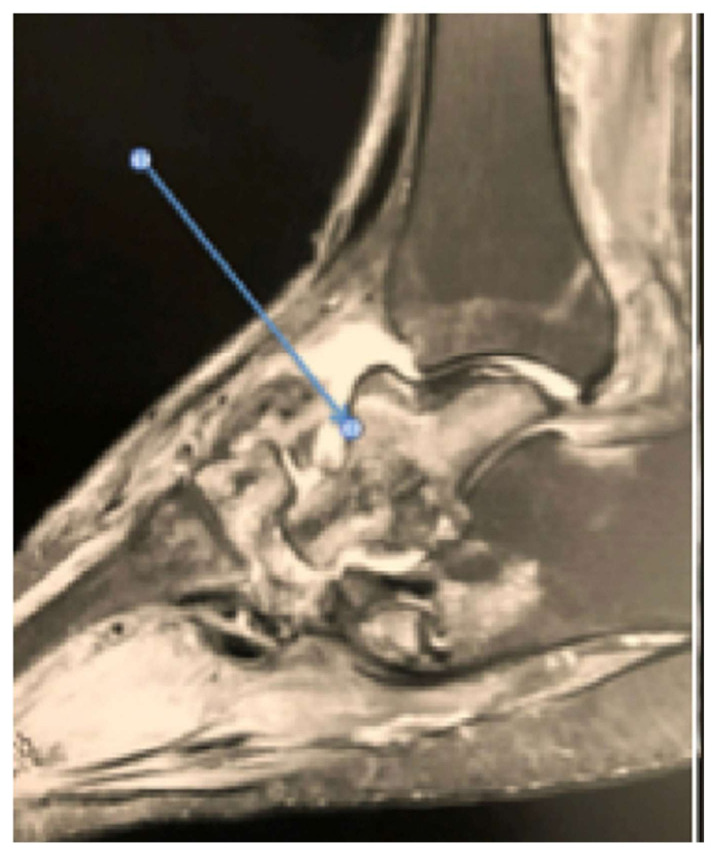
Magnetic resonance imaging at the time of diagnosis.

**Figure 3 medicina-58-00011-f003:**
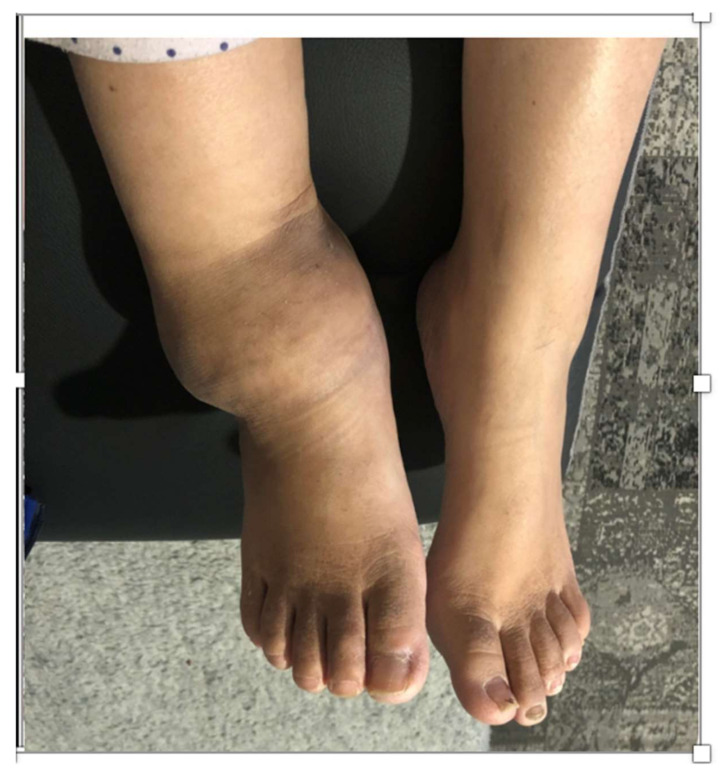
Clinical presentation of the untreated foot with Charcot neuroarthropathy.

**Figure 4 medicina-58-00011-f004:**
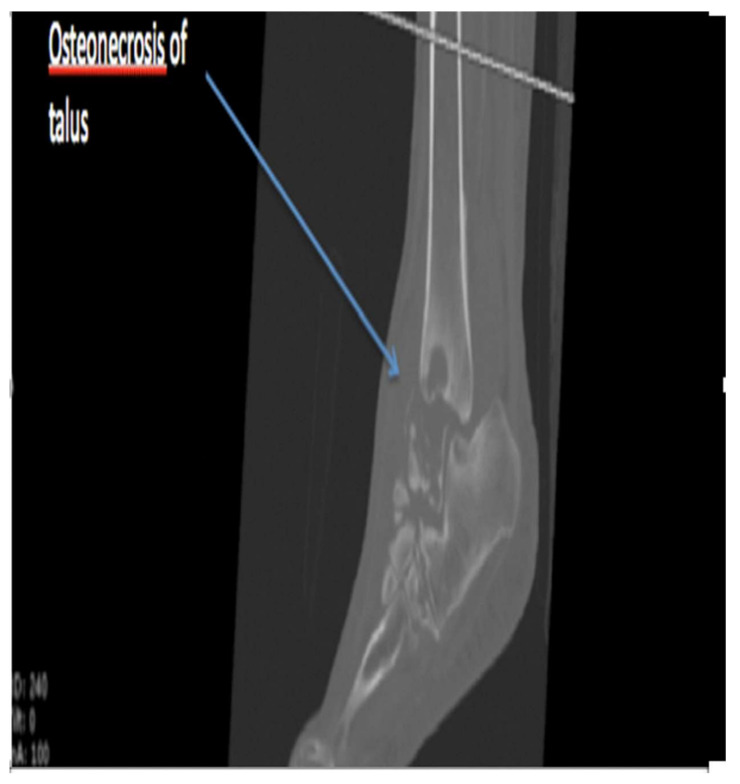
Computed tomography image showing osteonecrosis of the talus.

## Data Availability

The data that support the findings of this study are available from the corresponding author.

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
