# Peer review of "Fracture of the Bone Inducing Its Necrosis as the End Point in the Evolution of Untreated Neuroarthropathy"

_medicina, 2021, doi:10.3390/medicina58010011_

Round 1
Reviewer 1 Report
This case report demonstrates a case of a type 1 diabetes patient became a neuroarthropathy patient, lead to bone necrosis without treatment. The information of the case is detailed, and the dissusion is overall clear. however, there are still several concerns:
1. Considering the association of Charcot neuroarthropathy with diabetic neuropathy, does the patient have any relevant tests about her peripheral nervous condition?
2. The mechanism of CN are uncertain, beside inflammation, the others should be mentioned. like proprioceptive deficits secondary to peripheral neuropathy may result in ligamentous laxity, increased joint mobility, and injury due to minor trauma.
3. The discussion section mentions avascular (aseptic) necrosis and the different between avascular (aseptic) necrosis and CN should be described in detail, explaining the difference between the two different diagnoses for the treatment of the disease and other aspects.
4. Line 61 are labeled figure2 and mentioned contrast enhancement. however in Figure2, CT imaging seems to be non-equal scaled and not very clear, and it does not seem to be the enhanced CT imaging.
5. From the reader's point of view, the arrangement of both figure is very strange and does not match the order of the text. In the main text, the symptoms are stated first, and then the imaging changes are described.
6. The legends of 2 figures should be more detailed and accurate.
Author Response
1-Considering the association of Charcot neuroarthropathy with diabetic neuropathy, does the patient have any relevant tests about her peripheral nervous condition?
Response: Thank you for your comment. The diagnosis of peripheral neuropathy was confirmed by an abnormal monofilamant test and the tuning fork tests.
2-The mechanism of CN are uncertain, beside inflammation, the others should be mentioned. like proprioceptive deficits secondary to peripheral neuropathy may result in ligamentous laxity, increased joint mobility, and injury due to minor trauma.
Response: Thank you for your feedback. We opted to present the most recent hypothesis in the pathophysiology of CN (Jeffcotte) where ligament involvement is poorly described. However, we would like to add to this the amplification of ligament laxity in the modification of joint mobility and the presentation of trauma.
3-The discussion section mentions avascular (aseptic) necrosis and the different between avascular (aseptic) necrosis and CN should be described in detail, explaining the difference between the two different diagnoses for the treatment of the disease and other aspects.
Response: This difference in clinical symptoms was added to the text.
4- The addition of contrast enhancement showed significant fluid effusion in the talocrural joint and adjacent synovial thickening.
Response: Thank you for your remark. The sentence is indeed not very clear, so it was deleted.
- From the reader's point of view, the arrangement of both figure is very strange and does not match the order of the text. In the main text, the symptoms are stated first, and then the imaging changes are described.
- The legends of 2 figures should be more detailed and accurate.
Response: The figures have been changed and separated into clinical and radiological parts.
Reviewer 2 Report
I am grateful for the possibility to revise this case report.
Bone necrosis as the end point in the evolution of untreated neuroarthropathy is an interesting topic and may be a main focus of interest for readers.
The title is appropriate
The abstract sections reflect adequate the main objective of study
Introduction may be improved adding new information in order to provide an adequate state-of-the-art including some references. I suggest to include this references include in the attached to complete this requirement with regards to stiffness in diabetic population that authors do not included
doi: 10.1590/1806-9282.66.2.216
Figures and redaction of the results are presented in a correct way providing a good presentation of the main finding of the case repor
Discussion section was well-presented
However, authors should include some reference related to neuropathy. I suggest to include this reference to complete this requeriment on discussion section
doi: 10.1111/iwj.13263
Author Response
1-Introduction may be improved adding new information in order to provide an adequate state-of-the-art including some references. I suggest to include this references include in the attached to complete this requirement with regards to stiffness in diabetic population that authors do not included doi: 10.1590/1806-9282.66.2.216
Response: We added the reference as suggested by the reviewer.
Discussion section was well-presented. However, authors should include some reference related to neuropathy. I suggest to include this reference to complete this requeriment on discussion section
Response: Thank you for your suggestion. The proposed reference describes a predictive model to identify the risk of losing the protective sensibility of the foot in patients with diabetes mellitus. However, we believe that this link has little to do with the pathophysiology of Charcot neuroarthropathy.
Reviewer 3 Report
: title,
I think that fracture of talus resulted in necrosis. Therefore, neuroarthropathy lead to fracture without sensation.
Please change the title.
: At introduction, please describe unknown evidence on this case.
: At patients and methods, how did you diagnose Charcot neuroarthropathy?
Add the clinical diagnose criteria with reference.
: Line 65: please describe summary of the case with Neues
: Line 101: please add the review table on necrosis of bones associated with Charcot neuroarthropathy
Author Response
At introduction, please describe unknown evidence on this case.
Response: We added this information.
Please change the title.
Response: May we ask you to accept this title? The other reviewers did not object to the title, and we believe that it adequately conveys the subject of the paper.
t patients and methods, how did you diagnose Charcot neuroarthropathy?
Response: We diagnosed it with a clinical examination and an MRI (lines 47-50)
Line 101: please add the review table on necrosis of bones associated with Charcot neuroarthropathy
Response: We added the table as suggested.
Round 2
Reviewer 2 Report
Authors have addressed all my requeriments in the correct way, I believe it is suitable for publish
Author Response
Dear Reviewer
Many thanks for your comments
Kindly regards
Dr Dured Dardari
MD PhD
Reviewer 3 Report
Thank you for reply.
1: title. I think that fracture of talus resulted in necrosis. Therefore, neuroarthropathy lead to fracture without sensation, not bone necrosis.
2: Did you diagnose Charcot neuroarthropathy with a clinical examination and an MRI ?
There are no clear-cut set criteria. Therefore, you should consider some physical examination and image findings totally. See https://www.ncbi.nlm.nih.gov/pmc/articles/PMC6583154/.
3: please add the review table on necrosis of bones associated with Charcot neuroarthropathy
I can not find out the table..
Author Response
Dear Reviewer
Many thanks for your comments, here is my answers
1: title. I think that fracture of talus resulted in necrosis. Therefore, neuroarthropathy lead to fracture without sensation, not bone necrosis.
Dardari: the title has been changed according to your recommendation
2-Did you diagnose Charcot neuroarthropathy with a clinical examination and an MRI ?There are no clear-cut set criteria. Therefore, you should consider some physical examination and image findings totally. See https://www.ncbi.nlm.nih.gov/pmc/articles/PMC6583154/.
Dardari: Many thanks you for your comment, the diagnosis was initially clinical and as the patient was pregnant the confirmation of the Charcot neuropathy was carried out by the practice of the MRI, I am at your entire disposal to complete the information on the diagnosis
3-Please add the review table on necrosis of bones associated with Charcot neuroarthropathy
Dardari:Once again thank you for your comment I added between line 102 and 105 the necessary reference